# Price Promotions of E-Liquid Products Sold in Online Stores

**DOI:** 10.3390/ijerph19148870

**Published:** 2022-07-21

**Authors:** Shaoying Ma, Shuning Jiang, Meng Ling, Jian Chen, Ce Shang

**Affiliations:** 1Center for Tobacco Research, The Ohio State University Wexner Medical Center, Columbus, OH 43210, USA; shaoying.ma@osumc.edu; 2Department of Computer Science and Engineering, The Ohio State University, Columbus, OH 43210, USA; jiang.2126@buckeyemail.osu.edu (S.J.); ling.253@buckeyemail.osu.edu (M.L.); chen.8028@osu.edu (J.C.); 3Department of Internal Medicine, The Ohio State University Wexner Medical Center, Columbus, OH 43210, USA

**Keywords:** vaping, electronic cigarettes, e-cigarette, ENDS, e-liquid, e-juice, online store, tobacco marketing, price promotion, tobacco control

## Abstract

Background: Retailer price promotions are an important marketing strategy to attract consumers. However, there is scarce evidence on how retail price promotions are being implemented by e-cigarette online stores, particularly for e-liquid products that are not often found in brick-and-mortar stores and sold in lower prices compared to other types of e-cigarettes. Objectives: We collect data on e-liquid price-promotion activities from online stores using web scraping. From February to May 2021, we scraped the price promotion data of over 14,000 e-liquid products, from five popular online vape shops that sell nationwide in the US. We present descriptive analyses of price promotion on those products, assess price promotion practices in online stores, and discuss components of the final purchase price paid by online customers. Findings: Of the 14,000 e-liquid products and over, 13,326 (92.36%) were on sale, and each online store on average offered discounts from 9.20% to 47.53% for these products. The distribution of the after-discount price was largely similar across the five stores, and there is evidence that each store had adopted different price-promotion strategies. Conclusion: Despite low prices, price promotions are common for e-liquid products, which may undermine the effect of e-cigarette pricing policies such as excise tax that are designed to raise e-cigarette prices.

## 1. Introduction

In the United States, e-cigarettes are the most commonly used form of tobacco or nicotine products among youth and the most commonly used non-cigarette tobacco products among adults [1,2,3]. As an alternative product type to combustible tobacco, e-cigarettes are less harmful and may help smokers to quit smoking by making complete transitions to them [2]. However, e-cigarettes may also attract youth and young adults, [3], which has promoted many state and local governments to impose e-cigarette excise taxes [4,5].

Although taxes are very effective in reducing the demand for e-cigarettes, their impact could be offset by price promotions [6,7,8]. Thus far, there is no federal ban on promotional discounts applied to tobacco or nicotine products in the US [9], and nearly one in seven current adult e-cigarette users reported using price promotions during 2015–2016 [10]. Studies have further shown that price promotions are associated with e-cigarette use among both youth and adults in the US [11,12]. Therefore, it is important to generate evidence on e-cigarette price promotions to inform federal, state, and local governments about price-promotion restrictions, which have been underutilized to regulate e-cigarettes [13,14].

One important evidence gap is the form of price-promotion strategies by e-cigarette models (e.g., pods, cartridges, e-liquid, etc.). Existing evidence has predominantly focused on the pricing strategies of pods (e.g., JUUL) [15]. Although pod products arguably are the most popular e-cigarette model, the constantly evolving marketplace and changing regulatory environment (e.g., FDA rejected JUUL’s premarket tobacco product application) call for a comprehensive assessment of e-cigarette models that are beyond pod products.

Moreover, e-cigarette price promotions may take the forms of quantity discounts (e.g., bundles and buy X, get Y free) and direct price discounts (e.g., percentage off and coupons) [10,11,12,15,16,17,18,19]. Given the various designs of e-cigarettes, price-promotion activities could be different from those of traditional cigarettes and vary by e-cigarette models. For example, discounts on refillable cartridges or pods may rely more on quantity discounts rather than direct price discounts, whereas disposables and e-liquid bottles may rely mostly on direct price discounts; this could influence subsequent use behaviors. Furthermore, a previous study suggests that different e-cigarette models bear different tax burdens under the same tax structure (e.g., cartridges and pods bear lower tax burdens than bottled e-liquid when e-cigarette taxes are solely based on the volume of solution). Price-promotion policies, therefore, could be designed in a way to compensate different tax burdens resulting from the choices of tax structure [4].

The surveillance of e-cigarette price promotions further needs to expand to online sales spaces. According to various marketing statistics, about 15–33% of the total e-cigarettes sales during 2019–2020 were online and this percentage is projected to increase overtime [20]. However, existing surveillance work heavily relies on the Nielsen retail scanner data that primarily capture pod and cartridge products sold by large manufacturers in brick-and-mortar stores. It is, therefore, important to conduct surveillance of products sold online and their price-promotion activities, especially e-liquid products that are not well represented by the Nielsen retail scanner data.

The monitoring of online stores also has methodological advantages in capturing the price promotions of every single e-cigarette product on sale with great accuracy. In comparison, tobacco surveys measure price promotions based on participants’ responses that are subject to self-reported errors [9,10,12]. Promotional email tracking (e.g., Trinkets & Trash) is limited to a number of brands produced by large manufacturers. Store observation (e.g., V-STAR for point-of-sale assessment) is often limited to the overall assessment of ongoing promotions and impossible to capture the magnitude of price promotion for every single product on sale [21,22,23,24,25].

In this study, we collect price promotions for over 14,000 e-liquid products from five online stores, document the price-promotion strategies of each store, and assess price-promotion scale for every single e-liquid product. As such, this study fills in existing research gaps in monitoring online price promotions and provides future opportunities to compare price-promotion strategies by e-cigarette product models.

## 2. Materials and Methods

### 2.1. Selection of Online Vape Shops in Our Sample

We collected e-liquid price promotion data between February and May 2021 from five popular online vape shops that sell nationwide in the US. We selected the online stores based on Google search results and recommendations by vapers in an online forum, and obtained data of e-liquid products in each store using web data extraction. We searched for online stores on Google on 26 January 2021, using the search term “best online vaping stores in 2020.” It returned a list of “Best Online Vape Stores & Shops 2020” provided by vaping360.com (accessed on 26 January 2021). From that list, we selected the first three shops without physical addresses, since we sought online stores without physical shops that were reliant on online orders and mailing services to sell their vaping products to US customers. We also performed a Google search using the term ”best online vaping stores 2020 Reddit” on 26 January 2021 to obtain a list of recommended stores by vapers who were active in relevant Reddit discussions. We were able to find a 2020 discussion about “best online vape stores” on Reddit and selected two online stores with no physical addresses on their websites. Five national online vape shops comprise our sample; in order to mask store identities, we use stores from 1 to 5 to describe our observations and analyses.

### 2.2. Web Data Extraction

We scraped data of e-liquid product names as listed on webpages, pack size of each product (i.e., number of packs in one product), whether the product was bundled, actual price and original price, and whether the product was in stock. For all five stores in our sample, we obtained the following data: actual price, original price, in-stock status, and pack number. As for product bundling, stores 1, 4, and 5 sold some e-liquid products in bundles, and stores 2 and 3 did not sell any e-liquid products in bundles.

In this study, actual price, or after-discount price, refers to the sale price (the price paid by customers); original price, or before-discount price, refers to the list price (see Figure A1 in the Appendix A for an example). In Figure A1, we show how online stores display an original price and after-discount price for their e-liquid products. Specifically, we show a screenshot for an e-liquid product on each store website, taken on 6 August 2021. In each store, a straight line was written across the original price of an e-liquid product, and the after-discount price was highlighted. For instance, on the store 2’s website, the original price of “PINK LEMONADE-VAPETASIA–100 ML” at 6 mg was $29.99, and its actual price was $14.99. Using the data of the original price and actual price, we were able to calculate price promotion of e-liquid products in each store.

### 2.3. Calculation of Product–Level Price Promotion

We calculate product–level price promotion as a percentage discount (% off) as follows:price_promotion_ij_ = [(original_price_ij_ − actual_price_ij_)/original_price_ij_] × 100(1)

In the above equation, *i* denotes product and *j* denotes store. We obtain this product–level price promotion measure using original price and actual price (i.e., after-discount price) of each e-liquid product as listed on its product webpage. The product–level price promotion in our data ranges from 0% (no discount) to 96.50%. There are rare cases when the online shops labeled the original price of a product to be lower than its after-discount price; in those cases, we treat them as no discount (zero promotion).

### 2.4. Other Forms of Price Promotion

We also document the practices of e-liquid price promotion by the online stores in our sample, in addition to the calculation of price promotion in the form of percentage off based on the above equation. Specifically, we show that online vape shops could use sitewide promotion and/or free shipping for all e-liquid products during a limited period, on top of the percentage-off discount shown on each product webpage. We also present discounts sent via email by online vape shops to customers who had abandoned their shopping carts containing e-liquid products. Some of the stores in our sample adopted the strategy of product bundling, and priced two or more packs of the same e-liquid product as a bundle, or several different e-liquid products as a package. Furthermore, quantity discounts in the form of “buy x, get y free” were captured in store 1, but not in any of the other stores.

### 2.5. Costs of Purchasing Vaping Products from Online Stores

Lastly, we present the composition of costs for a customer to purchase a typical e-liquid product in each online store. Some of the components of the final price paid by an online shopper of e-liquid products were different from what they would be if purchased from physical stores. Our data provide relevant information for future regulations on marketing practices of online vape shops.

## 3. Results

In Table 1, we present the promotion strategies in each online store in our sample. For each individual e-liquid product in each online store, we calculate the price promotion in the form of percentage off based on Equation (1). Each of the five stores put at least some (if not all) of the e-liquid products on sale and offered a lower sale price relative to the list price on the webpage.

Four of the five stores used site-wide discounts. Specifically, one store offered “15% Off E-Liquid & CBD + Free Shipping” on 7 July 2021; one store offered “20% Off E-Liquids Sale” on 12 September 2021; one store offered “15% OFF All E-Liquids” on 12 September 2021 through 15 September 2021; store 5 sent coupons in email in July and August 2021, and offered discounts ranging from 25% to 35% off everything on their website, to incentivize potential customers who left their email addresses but did not complete check out for the e-liquid product (s) in their carts. Those sitewide-discounts applied to all e-liquids available on the store website for a limited period of time and at check-out. Therefore, when we calculated product-level promotion (i.e., percentage off), we did not capture such sitewide discounts.

Free shipping is another feature that is often seen as a promotion strategy. Store 1 offered free shipping for all e-liquid orders without a price limit on 7 July 2021. Four of the five stores in our sample also offered free shipping for orders that were above a certain dollar value. Based on their shipping policies as of 12 September 2021, store 1 offered “Free domestic shipping on orders over $65+”, store 2 had “Free Domestic US Shipping on Orders Above $80”, store 3 offered “Free Domestic US Shipping on Orders Above $75”, and store 5 had “orders over $100 (after any discounts) ship for free” in its shipping policy. Another important strategy is quantity discount or bundling. Three of the five online vape shops (stores 1, 4, and 5) employed the strategy of product bundling. Figure A2 presents the web data of product bundling that we scraped from February to May 2021. Store 1 was the only store that offered “buy x, get y free” discounts, captured on 28 October 2021.

In Table 2, we present the summary statistics of original price, actual price, and price promotion (as % off) in the whole sample and by store. In total, there are 14,477 e-liquid products from web data of five online stores. We have price data of 14,429 e-liquid products. When the actual price of a product is available, and its original price is missing, we treat the original price as equal to the actual price (i.e., zero promotion). We were thus able to calculate price promotion for 14,429 e-liquid products based on equation (1). Among them, 1103 products had either no promotion or a negative promotion in the raw data from the store websites; 1079 products had no discount, and were from three stores (1, 3, and 5); 24 e-liquid products had negative promotion, meaning the online stores labeled the original price to be lower than the after-discount price (12 from store 1, 12 from store 3). We replaced negative promotion with zero promotion in our analysis, treating negative promotion as no discount. There were 48 e-liquid products without price information in our data, all from store 4; this store website hid price information when products were not in stock.

Among the 14,477 e-liquid products in our sample, 9685 (66.90%) were in stock when we scraped them from store websites. Among the 14,429 e-liquid products that we obtained promotion data for, 9685 (67.12%) were in stock. The median promotional discount of those products was 40.93% off, and the average promotional discount was 38.89% off. The smallest discount was 0% off (i.e., actual price equals original price), and the maximum discount in our data of five online stores was 96.50%. Among the 9685 e-liquid products that were in stock and that we had promotion data for, the median and mean were similar: the median was 41.68% off, and the mean was 39.81% off. The average discount in store 5 was 47.53%, which was the greatest among the five stores. The average discount in store 1 was 9.20%, which was the smallest among the five stores.

In Figure 1, we present the distributions of e-liquid promotion, as well as after-discount price, by store using box plots. The median discount in store 1 is zero. The median discount in store 5 is 48.02%, which is the largest among the five stores. Store 3 has the greatest spread between its 75th percentile (50.13%) and its 25th percentile (35.22%); store 5 has the smallest interquartile range (IQR) among the five stores, and its 75th and 25th percentiles are 52.02% and 44.02%, respectively.

We also show in Appendix A
Figure A2 the percentage of e-liquid products bundled in each of three stores (1, 4, and 5): 2.58%, 5.67%, and 24.30% of the e-liquid products were sold in bundles by stores 1, 5, and 4, respectively. The other two stores in our sample did not sell any e-liquid products in bundles. Appendix A
Figure A2 also shows the frequencies of pack number in each of the five stores in our sample. In store 1, out of 1669 e-liquid products, 132 were in two packs, and the rest were in single packs. Store 5 had most of its e-liquid products in single packs as well; among its 1868 e-liquid products, 238 were in two packs, and 31 of them were in packs of three or more. Among the 3560 e-liquid products sold by store 4, 2678 were single packs, 180 products were in two packs, 347 products were in three packs, and the rest were in four or more packs. At store 2, 2735 out of the 2803 e-liquid products were in single packs, and the remaining 68 were in two packs. Store 3 had a total of 4577 e-liquid products on its website; among them, 4156 were in single packs, 410 were in two packs, and 11 were in three or more packs.

We also document the components of the final price of a typical e-liquid product when ordering from each online store (see Table A1 in the Appendix A for more details). In Table A1, we take products shipped to Minnesota as an example, and Minnesota is the first state to impose e-cigarette excise tax in 2010. The e-cigarette excise tax rate in Minnesota is 95% of the wholesale price. As shown in Table A1, for a shipping address in Minnesota, a customer would pay the sum of sales price, shipping cost, adult signature fee, Minnesota state tax, county tax, city tax, special tax, and Minnesota state vapor tax, for ordering an e-liquid product online from store 1. If they were to order from store 2, they pay the sum of sales price, adult signature fee, shipping cost, Minnesota sales tax, and excise tax. Store 3 charges the sum of sales price, adult signature fee, shipping insurance, and Minnesota excise tax, and sales tax, and it offers free shipping. Store 4 charges the sum of sales price, shipping cost, and adult signature fee for an e-liquid product shipped to Minnesota. If the customer were to order from store 5, they would pay the sum of the sales price, bottle surcharge, shipping cost (signature required), and sales taxes.

The screenshots of the checkout webpages of the various stores were taken on the following dates: 10 May 2021 (store 1); 11 May 2021 (stores 2 and 4); 29 June 2021 (store 3); and 19 July 2021 (store 5). As shown in Table 1 and Table 2, Figure 1 and Figure A1, and Table A1, e-liquid products in online stores are very affordable, and shipping costs (including adult signature fee), as well as e-cigarette excise state taxes (when stores comply with taxation), could constitute substantive portions of the final purchase cost for customers who shop online. Online vaping stores could use price promotions and/or offer free shipping to offset the effect of e-cigarette excise taxes imposed by states and local jurisdictions.

In Table 3, we estimate the differences in product-level promotional discounts, as well as standardized e-liquid prices across five stores. Each of the four stores 2–5 offered on average a higher product-level promotional discount and lower e-liquid price, relative to store 1.

## 4. Discussion and Conclusions

Our study presents a snapshot of e-liquid price promotion activities for over 14,000 e-liquid products in five popular online vape shops that sell nationwide in the U.S. We found that over 92% (92.36%) of the e-liquid products had promotional discounts, and on average each store offered between a 9.20% and 47.53% discount for e-liquid products. Among the five stores, two offered promotional discounts for every e-liquid product on their websites; e-liquid price promotions in store 2 ranged from 12% to 60.03% off, whereas store 4 offered between 17.66% and 96.50% off. Given that online stores do not pay for expenses associated with a physical location and may have lower operational costs than brick-and-mortar stores, price promotions may further boost their competitiveness in the market. Indeed, our prior study shows that the average price of e-liquid sold in online stores was lower than the prices reported in surveys and sales data [26].

According to the International Tobacco Control (ITC) surveys, 23% of current adult e-cigarette users and 14–16% of youth and young adult e-cigarette users reported purchasing products from online stores [27,28]. As online sales of e-cigarettes with widely-available price promotions, online stores will likely remain as a major source for consumers to obtain cheap e-cigarette products [20]. We further document that multiple price promotion forms are used by online stores, including quantity discounts that encourage consumers to buy multiple products or multiple units of a product at a discounted price, which will subsequently impact use patterns. Therefore, it becomes important for states and localities to consider price-promotion restrictions (e.g., establishing price floors) in addition to taxation, if the policy goal is to increase the costs of e-cigarette products. The US Food and Drug Administration (FDA) could also regulate marketing behaviors that involve price promotions such as banning coupons to reduce the availability of very cheap products.

Our data also fill in the research gap of monitoring price-promotion activities of e-liquid products—a model not well captured by existing data sources such as the Nielsen retail scanner data. We demonstrate the sizable discount on e-liquid products is due to price-promotion activities, suggesting that e-cigarette taxes would be offset by these promotions. However, since e-liquid products bear a higher tax burden compared to pod and cartridge products when e-cigarettes are taxed based on volume sizes (pods and cartridges have a much smaller volume size compared to bottled e-liquid), this offset may reduce the tax burden difference by product models. Future studies should evaluate whether this is the case and if so, whether this is a desirable policy outcome that would promote public health.

Our data and findings further offer an opportunity to monitor online sales when the regulatory environment continues to change. On 23 June 2022, the US FDA issued marketing denial orders (MDOs) to JUUL e-cigarette products [29], which could promote the sales of alternative e-cigarette products including e-liquid products online. The proposed flavor restrictions that would ban charactering flavors in combustible tobacco could have a similar impact to incentivize the sales of e-liquid products with a wider range of flavor profiles. Monitoring price promotions will allow us to ascertain whether e-liquid manufacturers will capitalize on these opportunities and use price promotions to increase their market shares. On the other hand, it is unclear whether these e-liquid products sold online have submitted premarket tobacco product applications (PMTA) required by the FDA. It is possible that price promotions are a strategy to sell existing stocks of e-liquid products before they are outlawed by the FDA enforcement. We will assess the product availability overtime in future studies to elucidate promotion strategies.

Our study has several limitations. First, it focuses on price promotions of e-liquids only and does not include other product types such as open-system devices and disposable e-cigarettes such as JUUL. Second, we used a convenience sampling strategy and selected online stores based on simple Google and Reddit search results. Nonetheless, we did not intend to collect representative samples of online stores. Finally, stores may use promotions as a strategy to attract consumers without actually lowering costs, which cannot be discerned in our approach. Future studies that follow price-promotion activities over time will be able to ascertain whether price promotions claimed by the stores indeed lower costs (e.g., to observe prices with and without claimed promotions).

Despite the limitations, our study is the first in-depth investigation of price-promotion activities in the online marketplace. The evidence suggests that online stores use common price promotion forms such as quantity discounts to encourage purchases. Policymakers may need to impose price-promotion restrictions, in addition to taxes, to raise product prices. Nonetheless, there are also opportunities for federal, state, and local regulators to utilize price-promotion restrictions in combination with taxes to promote e-cigarette product types that have greater public health benefits than costs (e.g., imposing more restrictions on models preferred by youth and young adults). Innovations in policy design and enforcement will be needed to address these challenges.

## Figures and Tables

**Figure 1 ijerph-19-08870-f001:**
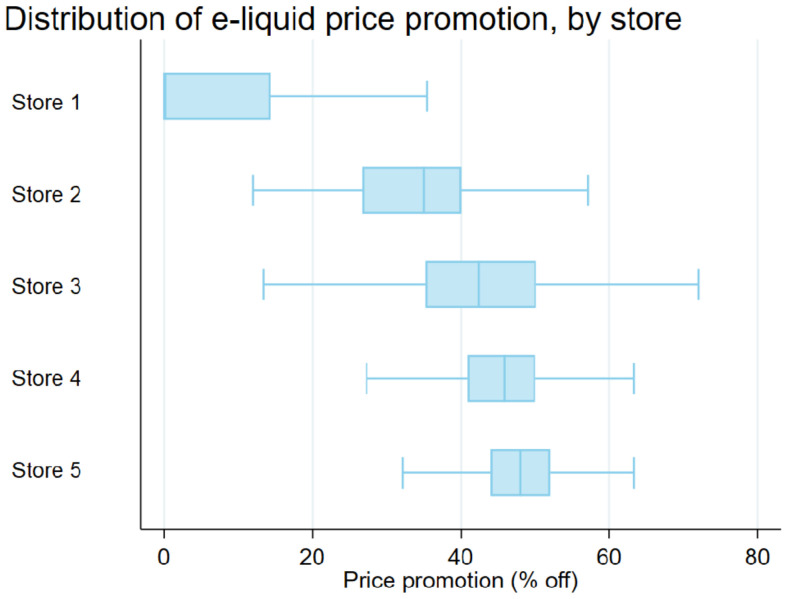
Distribution of e-liquid price promotion by store, in box plots (*excluding outside values*).

**Table 1 ijerph-19-08870-t001:** Promotion strategies in each online vape shop.

Promotion Strategies	Stores
Sales (Percentage Off) ^1^	Stores 1, 2, 3, 4, 5
Sitewide Discount ^2^	Stores 1, 2, 3, 5
Free Shipping ^2^	Stores 1, 2, 3, 5
Product Bundling ^1^	Stores 1, 4, 5
Buy X, Get Y Free ^3^	Store 1

^1^ Price promotion in the form of percentage off was calculated for each e-liquid product sold in each of the five stores in our sample; calculation was based on equation (1), and the data we used for those calculations were scraped from store websites from February to May 2021; we also obtained data of bundled products from those web data. ^2^ We checked sitewide discounts as well as shipping policies on store websites at two time points, 7 July, and 12 September 2021; we also documented the coupons that we received by email from store 5 during July–August 2021, by leaving an email address at the checkout webpage. ^3^ Quantity discounts in the form of “buy x, get y free” were not found in any of the five stores when we initially scraped their web data during February–May 2021; later, on 28 October 2021, we documented the only store that offered “buy x, get y free” discounts on some of its e-liquid products.

**Table 2 ijerph-19-08870-t002:** Summary statistics.

	*n*	Mean	s.d.	Min	Max
*Panel A. Whole Sample*					
Original Price ($)	14,429	27.33	19.28	3.99	287.88
Actual Price ($)	14,429	15.81	9.19	0.49	131.89
Price Promotion (% off)	14,429	38.89	16.10	0	96.50
*Panel B. Store 1*					
Original Price ($)	1669	19.68	7.70	3.99	99.75
Actual Price ($)	1669	17.57	7.09	3.49	89.95
Price Promotion (% off)	1669	9.20	15.69	0	80.92
*Panel C. Store 2*					
Original Price ($)	2803	23.08	3.85	16.99	34.99
Actual Price ($)	2803	15.06	2.13	7.99	23.99
Price Promotion (% off)	2803	33.92	8.53	12	60.03
*Panel D. Store 3*					
Original Price ($)	4577	23.37	4.27	4.99	39.99
Actual Price ($)	4577	12.84	2.87	2.49	27.99
Price Promotion (% off)	4577	43.98	12.54	0	85.32
*Panel E. Store 4*					
Original Price ($)	3512	39.36	34.57	10.99	287.88
Actual Price ($)	3512	20.29	16.05	0.49	131.89
Price Promotion (% off)	3512	45.73	8.41	17.66	96.50
*Panel F. Store 5*					
Original Price ($)	1868	27.64	10.63	7.99	99.99
Actual Price ($)	1868	14.17	5.54	5.99	74.99
Price Promotion (% off)	1868	47.53	10.22	0	75.01

**Table 3 ijerph-19-08870-t003:** Price and price promotion differences across stores.

Dependent Variable	Promotion	Log(Standardized Price)
Store 2	24.614 ***	−0.072 ***
	(0.345)	(0.015)
Store 3	34.674 ***	−0.445 ***
	(0.319)	(0.014)
Store 4	36.424 ***	−0.356 ***
	(0.331)	(0.015)
Store 5	38.219 ***	−0.388 ***
	(0.375)	(0.017)
N	14,409	14,409

Notes: The dependent variable **promotion** is the product-level discount as percentage off (e.g., 0 means no discount, and 20 means 20% off); in the third column, e-liquid price is standardized as U.S. cents per milliliter; the above cross-store differences are estimated with store 1 as the reference group; *** statistically significant at 1% level.

## Data Availability

The price data will be made available through another publication in the review. We will also publish the data with this study.

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
