# Peer review of "Price Promotions of E-Liquid Products Sold in Online Stores"

_ijerph, 2022, doi:10.3390/ijerph19148870_

Round 1
Reviewer 1 Report
Abstract should be rewritten to indicate the significance of the topic, research methodology, and major findings of the study.
Introduction also needs a serious revision. The authors should write what their motivation is and what the novelty of the study is.
The authors should present a literature including similar studies, because many sellers use the price promotions.
I don’t understand what the authors tried to do in this paper. Why they study this topic. They should make a connection between article and the journal.
The study needs a further statistical analysis for publication in the Journal.
There is a serious problem in use of tenses in the study.
Author Response
- Comments to the Author
English language and style: Extensive editing of English language and style required.
Response: Thank you for this comment. A copy editor has reviewed our manuscript and made edits.
- Comment: Abstract should be rewritten to indicate the significance of the topic, research methodology, and major findings of the study.
Response: Thank you for this suggestion. We have revised our abstract.
- Comment: Introduction also needs a serious revision. The authors should write what their motivation is and what the novelty of the study is.
Response: Thank you for your suggestion. We have revised the introduction section to better motivate our study and highlight its novelty.
- Comment: The authors should present a literature including similar studies, because many sellers use the price promotions.
Response: Thank you for pointing this out. We revised the introduction section and discussed more extensively the relevant literature on price promotions, as well as the contributions that our study makes.
- Comment: I don’t understand what the authors tried to do in this paper. Why they study this topic. They should make a connection between article and the journal.
Response: We have significantly revised abstract, introduction, and conclusion. We hope this revision will make the article more clear.
- Comment: The study needs a further statistical analysis for publication in the Journal.
Response: We have added statistical analysis to test the differences in product-level promotional discounts and e-liquid standardized prices (US cents per milliliter) across five stores in our sample.
- Comment: There is a serious problem in use of tenses in the study.
Response: Thank you for pointing this out. A copy editor has reviewed our manuscript and made edits.
Reviewer 2 Report
The study aim is interesting, but this study repeats well-known and well-measured facts. There is a limited novelty in this study. The use of web and price promotion via internet was discussed in numerous papers.
Moreover, the authors used multiple overwhelming sentences without sufficient reference (e.g. lines 39-40).
The study design should be more precise and base on scientific evidence.
Methods used for "Calculation of Price Promotion" are unclear.
The research is limited to some groups of products, but the US market is much more extended and such products like JUUL and other vaping devices may have a different approach. Moreover, there is a difference between e-liquid and e-cig devices. The problem is more complicated and complex rather than simple analysis presented by the Authors.
Author Response
- Comments to the Author
The study aim is interesting, but this study repeats well-known and well-measured facts. There is a limited novelty in this study. The use of web and price promotion via internet was discussed in numerous papers.
Response: Thank you for this comment. We believe the novelty of our study is in that we are the first to obtain and analyze product-level data from online vape shops that sell numerous brands of e-cigarettes and describe the price promotion activities in those stores. We have added a paragraph to the discussion section to talk about what’s already in the literature as well as what contributions our study makes.
- Comment: Moreover, the authors used multiple overwhelming sentences without sufficient reference (e.g. lines 39-40).
Response: Thank you for this comment. We have added citations in the introduction section as well as the discussion section.
- Comment: The study design should be more precise and base on scientific evidence.
Response: Thank you we have added references and rationale of conducting this study.
- Comment: Methods used for "Calculation of Price Promotion" are unclear.
Response: We modified this section to be more clear that this price promotion measure is at product level. In addition, we mentioned that the original price and after-discount price (i.e., actual price) were scraped from each e-liquid product’s webpage at each online store, and Figure A1 presents screenshots to demonstrate how each store lists original price vs. after-discount price on product webpage.
- Comment: The research is limited to some groups of products, but the US market is much more extended and such products like JUUL and other vaping devices may have a different approach. Moreover, there is a difference between e-liquid and e-cig devices. The problem is more complicated and complex rather than simple analysis presented by the Authors.
Response: This is a great point. We discussed that one of the limitations of our study is that the data were limited to e-liquids, and did not include other e-cigarette product types such as devices. As we continued our web scraping efforts, we have obtained web data of other product types including disposables, devices, etc. Those data are in the process of being cleaned, and we plan to assess those other products in our future studies.
Round 2
Reviewer 1 Report
Dear Authors;
You have already made significant revision, the conclusion section needs a serious improvement considering the discussion and related literature.
Also you should give suggestions about future studies.
Reviewer 2 Report
The authors address some of the comments, but major concerns still exist:
- study design is unclear (see previous round comments 3 and 4)
- the most famous e-cigarette (JUUL) was missed
- the Authors mentioned that they will collect additional data (e.g., on JUUL) but these data will be presented in the further publications
- there is a limited novelty in this study, even after some minor changes applied by the authors (the core problems still exist - major products are missed; findings are very limited and mostly in line with well-known and well-documented facts)
